# The Robben Island Diversity Experience: It’s about the Journey, Not the RIDE

**DOI:** 10.3390/ijerph19159064

**Published:** 2022-07-25

**Authors:** Michelle S. May

**Affiliations:** Department of Industrial and Organisational Psychology, School of Management Sciences, College of Economic and Management Sciences, Mucklenuek Campus, University of South Africa, Pretoria 0002, South Africa; mayms@unisa.ac.za

**Keywords:** diversity dynamics, container, analytic autoethnography, race, gender, position, socio-economic conditions, denigration, idealisation, integration

## Abstract

The Robben Island diversity experience (RIDE), a conference using a group relations training model, was held annually from 2000 to 2013, (with RIDE not taking place in 2004). During RIDE, underlying, unconscious, and covert South African diversity dynamics are studied as they manifest among managers and officials in the fields of change, diversity management, transformation, and human resources management. The participants (staff and membership) of the conference collectively form a temporary organisation in which they explore unconscious dynamics and processes as they relate to dealing with diversity challenges. A qualitative, analytic auto-ethnographic research approach was used to elicit data. The data consisted of different sets of process notes compiled during my involvement in RIDE (2000 to 2014). Thematic analysis was used to analyse and interpret the data to allow insights into diversity dynamics. The findings suggest that in the early events RIDE citizens displayed a preoccupation with race which was expressed through attempts to re-organise the race hierarchy in the unconscious, and the need to form good-enough relationships with old enemies through searching for the absent white-man. In more recent years it became evident that groups are reluctant to explore differences of organisational position and socio-economic status in racial sameness. The experimentation with leadership within and across race groups and gender was a preoccupation of members in a more recent RIDE. Further, the marginalisation of women, through the intersection between diversity characteristics, also became apparent. It seems that through RIDE as a container, the consultants as containers, and the willingness of the participants to engage with their diversity challenges, more diversity dynamics have become accessible to the RIDE citizens. This understanding of diversity dynamics can be used in South African and international organisations by researchers, consultants, and managers to enhance their understanding of diversity dynamics in organisations and inform the nature of the different diversity management initiatives implemented in these organisations.

## 1. Introduction

When working with diversity and diversity management our focus is usually on the conscious factors (factors within our awareness) which drive our observed behaviour. Much research has been done on the conscious manifestations of diversity in organisations and how diversity management can be implemented to ensure to increase the benefits and reduce the disadvantages of diversity for all stakeholders in and of organisations [1]. However, by focussing on the influence of unconscious factors (factors outside of our awareness) on diversity will assist in forming a more holistic understanding of observed human behaviour in the context of diversity and diversity management [1,2,3]. Several research projects have focused on the experiences of diversity in group relations conference from the perspective of the researchers, consultancy staff, the directors and participants of such conferences [4,5,6,7]. Ref. [8] explored the experiences of the directors of a group relations conference (RIDE), whereas [6] explored the person-in-role implications for the director of a group relations conference exploring diversity. Research about the dynamics in group relations conferences exploring diversity include, but not limited to, the Nazareth Group Relations conferences, which were held on six occasions, initially focused on Germans and Jews and later included Palestinians and others [4,5]. The research about this conference reported on the unconscious dynamics evident when studying hatred and the historical roles of Germans and Jews in the Holocaust. The conference, entitled Authority, Leadership and Peacemaking, emphasized the role of Jewish, Palestinian and other Arab Diaspora communities in the Israeli–Palestinian conflict [7]. Although this research about RIDE differed from the aforementioned research, viz. RIDE was held consecutively from 2000 to 2003 and 2005 to 2015, and focused on diversity dynamics primarily in South African organisations and society. The middle-term nature of RIDE allowed me to reflect on diversity dynamics spanning over a 13-year period.

The Robben Island diversity experiences (RIDE) is a group relations training event using Tavistockian Technology (also known as the Group Relations training model) to study the underlying, unconscious, and covert South African diversity dynamics as they manifest among managers and officials in the fields of change, diversity management, transformation, and human resources management. These diversity dynamics evident in RIDE may also be viewed as a microcosm of South African diversity dynamics since 2000. The Robben Island Diversity Experience has contributed to the understanding about individual and group diversity dynamics. Diversity dynamics is defined as a relational phenomenon through which individuals, across differences and similarities, make certain assumptions about others. The individuals then behave, on both a conscious and an unconscious level, in a particular way towards each other, based on these assumed differences and similarities [1,8,9].

This article is an exploration of my reflections about diversity dynamics. My aim is to describe the experiences of a few hundred people participating in the Robben Island Diversity Experience (known as RIDE) from 2000 to 2013 (with RIDE not taking place in 2004), using analytic autoethnography to analyse and interpret the conscious and unconscious diversity dynamics operating among these participants.

In sharing with you my reflections about the diversity dynamics operating in RIDE, I start by explaining the significance of Robben Island as a container for South African diversity.

## 2. Literature Review

### 2.1. Robben Island: Container for Diversity

Through the ages society has showed us a particular way of dealing with those who at times seem to be different from us: by banishing them and/or their leaders to another continent (for example Africa, Australia, America) or to islands (for example Alcatraz; Robben Island) [8]. In exploring the history of Robben Island, it seems that, at first, the island represented the banishing of leaders from South Africa: Autshumato, a Khoi leader, was the first political prisoner exiled to Robben Island in 1658 because he was taking back cattle the people believed to have been unfairly confiscated by Dutch settlers; Autshumato was also one of the few prisoners to have escaped successfully from Robben Island. During the 1818–1819 Frontier War, one of the many fought in that century between the British and the Xhosa about land, a freedom fighter, Makana, surrendered in an attempt to ensure peaceful negotiation. He and 30 other warriors were sent to Robben Island and in 1820 he and the 30 men attempted to escape from the island but did not survive. Over time, 3000 black male activists, who included Robert Sobukwe, Nelson Mandela, Kgalema Motlanthe, and Jacob Zuma were imprisoned on Robben Island because of their resistance to the apartheid system. A black freedom fighter not only creates anxiety, but also confronts the idea of white supremacy [10].

Other people that the island contained were the chronically sick—a leper colony was established in 1845—and the mentally ill—a women’s asylum was established in the middle of the 19th century. Religious leaders were also banished to the island. The Moturu kramat was built in 1969 to commemorate Sayed Abdurahman Moturi, one of Cape Town’s first imams, who was exiled to the island in the mid-1740s and died there in 1754. The existence of these groups could be denied by thrusting them out of consciousness onto an island separated from the mainland by 14 km of seawater [10].

By banishing these freedom fighters, the chronically sick and mentally ill to Robben Island we protect ourselves from finding the skill and doing the work required to deal appropriately with the conflict which arises with diversities. Another aspect of banning the other could be linked to scarce resources. Instead of negotiating the equitable distribution of scarce resources among groups [11], the leaders of the dominant group could banish the leaders of the other groups to an island to avoid any challenges with regard to distribution of resources [10].

In the previous dispensation in South Africa, Robben Island became the perfect dumping ground for those in power—a place where elements that became anxiety-provoking could be pushed out of awareness. More than this, it also became a place where impurities, such as ideologies and beliefs could be contained. For example, aspects of the apartheid ideology were contained on Robben Island. Robben Island also became one of the symbols of the Struggle for black emancipation, embodying the conflict between the apartheid government and freedom fighters [10].

With the release of the last political prisoners in 1991, and especially the release of Nelson Mandela, and the 1994 election, Robben Island has become:A symbol and celebration of freedom;A place of pilgrimage—the pilgrimage of freedom fighters and the pilgrimage or visit of tourists (national and international) due to its world heritage site status and its symbolic significance to the world;A place of celebrating connections, for example, each year 14 February couples are allowed to marry on Robben Island;A container for the Robben Island Diversity Experience (since 2000).

A rock pile at the entrance of the lime quarry where Mandela and other struggle heroes worked, was started by Mandela after he spoke at a 1995 reunion of over 1000 ex-political prisoners gathered together again on Robben Island. As he left the quarry where the prisoners had toiled, he put down a stone. Thereafter, each man picked up a small rock or stone in the quarry area, and they placed them in a pile to stand as a simple memorial to their years of hardship and struggle on the one hand, and their liberation on the other. Based on the above, it is evident that Robben Island is both container and contained. At one time in history, it contained the denigrated parts and now it contains the idealised parts, not only for South Africa, but also to some extent for the world. The island serves as a symbol of both past denigration and current liberation and hope with regard to South African diversity dynamics [1,10].

### 2.2. RIDE: Container of Diversity Dynamics

To hold on to the aspects of the old and to use or re-integrate the banished aspects, we need to deal with our differences and similarities, firstly as these were construed by the previous dispensation, before working towards celebrating our diversity. Diversity has become one of the cornerstones of South African life. This phenomenon can perhaps best be understood by studying the underlying human dynamics referring to relatedness between people, the identity of those involved, the role of power in these relationships, and the effect of one’s reference systems in understanding diversity [2].

Professor Frans Cilliers and I were asked by the directors of an independent consultancy firm, Marius Pretorius and Derek Hendrikz, to design and present a group relations training event to study the South African diversity dynamics as they manifest among managers and officials in the fields of change, diversity management, transformation, and human resources management, as well as interested citizens [8]. In 2000 and 2001, I took up the role as associate director and 2002 until 2013 as director of RIDE (with RIDE not taking place in 2004).

The island was chosen by the hosting organisation for its marketing attraction, as well as the historical and symbolic significance of Robben Island pertaining to South African diversity dynamics. The first RIDE was held in 2000. A general invitation to attend RIDE is annually sent out too many organisations in South Africa, specifically to managers of corporate diversity and transformation programmes. Information about and an invitation to the event also reached other interested people, in South Africa, southern Africa, and abroad. We also shared research about RIDE nationally and internationally since 2000. On average, 35 participants attended each event since 2000—with one event attended by as many as 80 and another by as few as 20 participants. The gender and race ratios seemed to be proportional [8].

### 2.3. RIDE: Group Relations Training Event

A group relations conference is different from the traditional understanding of what a conference is. The participants (staff and membership) of the conference collectively form a temporary organisation in which they explore unconscious dynamics and processes as they relate to dealing with diversity challenges (see 12, 4, 13]. The programme consists of various large, small, inter-group and institutional events, as well as reflective events [8]. The staff of RIDE provides the members with an opportunity to learn about diversity dynamics in the context of interpersonal, intergroup and institutional relations within the conference institution. However, what members learn about the diversity dynamics lies within the members’ own authority. Thus, through the group relations training model, the link between unconscious dynamics created within groups, and between individuals and groups, as well as the impact of existing and emerging systemic dynamics, can be explored in the group relations conference [12].

### 2.4. Primary Task of RIDE

The primary task of RIDE is to provide opportunities for participants to study South African diversity dynamics as they unfold in the “here and now” of the event, in order to understand how participants, perceive, interpret, and act towards individual and collective diversity. By working to the primary task of RIDE, members can begin to work on their own diversity-related challenges and examine ways in which they interact with and contribute to their diversity dynamics in the different spheres of their lives (RIDE manual). Participants (staff and members) should also eventually [3,8].

### 2.5. Resources

The consultants take responsibility and authority to provide the boundary conditions of task, space (territory), and time, in which the underlying processes of the temporary organisation can take place so that all participants can engage with the primary task [13,14]. The purpose of these boundaries is to provide a psychologically safe environment, or differently put, a containing environment in which learning about diversity dynamics can occur. These boundaries are:*Territory*. This is the venue at which the conference is held. Here, the territory is not only the conference venues, but also Robben Island itself;*Time.* The staff adheres rigidly to time boundaries as indicated on the conference programme. For example, consultants will get up and leave a session without participating in the usual social niceties;*Task.* Within the conference-as-a-whole and each of the conference events, staff and members work towards a particular task, all of which are linked to the primary task;*Consultation.* The group relations training consultant is actively involved in the conference, formulating working hypotheses, and interpreting behaviour processes and dynamics in the here-and-now, on the basis of his/her own observations, experience and expertise, as well awareness of on diversity dynamics [15,16,17].

### 2.6. The Director during the Group Relations Event

Prof Frans Cilliers was director of RIDE in 2000 and 2001. I have been director from 2002 until 2013, with RIDE not taking place in 2004. In 2006, a conference was held in June and November. In June 2006, Dr Lerato Motswoaledi took up the role of director of RIDE and in November 2006 I was the director [8]. I am coloured, black, African woman who grew up as part of a middle-class family in a rural town, Genadendal (Valley of Grace), one-and-a-half-hour drive from Cape Town. We then moved to a town, from where we (when I was a pre-teen) moved to Cape Town. I was (passively) part of the political struggle during my high school years and to some extent during my tertiary education. I studied at two historically white universities and obtained my doctorate, in which I used a system psychodynamics to understand the dynamics in a historically black university, from the University of South Africa. My first job as a lecturer was at a historically black university in Pretoria, after which I worked at UNISA until now. I received training in the group relations model and systems psychodynamic stance nationally and internationally and worked in different roles as part of staff in national and international group relations conferences. In the national conferences, I worked with themes pertaining to diversity. This and my own diversity characteristics, prepared me to direct a conference which explored diversity dynamics.

The director, besides designing the conference, provides appropriate (emotional) containment for staff and members (negative capability), and manages the boundary conditions (positive capability) as suggested by [18,19] from the first planning meeting for RIDE until a debriefing session with staff after the event. The concepts of container and containment will be discussed in relation to the role of the consultants.

### 2.7. Consultants during a Group Relations Training Event

Ref. [20], p. 102 describes the task of the consultant as being to confront the members, without affronting its members; to draw attention to group behaviour and not to individual behaviour, to point out how the group uses individuals to express its own emotions, how it exploits some members so that others can absolve themselves from the responsibility for such expression.

The consultants do not have a facilitative role but consult to the unconscious processes of the group as they happen in the here-and-now as staff and group members interact to complete a task. Although the consultants provide clear direction on what the primary task is, no instructions are provided as to how to achieve the primary task or what to do. The consultants do not answer the members directly or structure the conversation for them [21]. The consultancy provided is not about a specific group member, but rather about how a specific group member is used to further the group’s underlying, unconscious processes. This understanding is then used to provide membership with interpretations and working hypotheses about what is happening in the group, across groups, and so on. Membership has the authority to accept those interpretations and working hypotheses that prove to be valid, and to reject what is not. Through this process, group members take full authority for the content of the discussions. They can reconsider the way they work with each other and other groups, and how to change or maintain their respective positions [22]. 

The consultants have to maintain nurturing in the face of rage, envy, and jealousy that can arise when the members experience frustration, apprehension, fear, and loss when they have to learn about diversity dynamics in a group relations conference. Although positive capability consultants provide boundary conditions (venue, time, task, space and consultancy) and structuring functions to create a safe, holding environment in which members can learn about specific topics [23]. Although positive capability is necessary, it does not provide sufficient containment because it does not allow the consultant to deal with the overwhelming, often unconscious, feelings arising from learning about diversity dynamics. Therefore, negative capability should be present in and used by the consultant [24]. Negative capability refers to the capacity of being in uncertainties, mysteries, and doubts, without any irritable reaching after fact and reason [24]. The consultant tolerates confusion and unknowing when members experience frustration—not taking away the frustration, but sitting with the frustrating elements in mind [25]. The consultant keeps these elements in mind and digests or detoxifies them through a (psychological) metabolic, disentangling process in the self (these are Bion’s terms) making sense of these elements, i.e., thinking about it and in so doing providing containment for the member [26,27,28]. Thus, the consultant demonstrates, through calm receptivity and maintaining a self-reflecting stance, to the members that anxiety-provoking aspects can be understood, thought about and tolerated [27,29]. Through this process the members develop their own capacity for reflecting on anxiety-provoking aspects [29,30] pertaining to diversity dynamics. Thus, the members can, one hopes, move beyond not merely spitting out or projecting into or onto the other their unresolved diversity challenges. This is what is known as the consultant being a good-enough container. In a similar vein, RIDE serves as a container in which staff and members can work to the primary task, and Robben Island acts as a container for the work we do (participants) as illustrated in Figure 1.

## 3. Research Methodology

Qualitative and descriptive research [31] was chosen to allow for in-depth exploration of diversity dynamics in a specific experiential diversity experience [32]. Hermeneutic phenomenology [33] was chosen as research paradigm, which allowed for the in-depth understanding of the participants’ experiences around diversity. The paradigm also enabled us to interpret the data from the systems psychodynamic stance in an attempt to develop knowledge around the conscious and unconscious manifestation of diversity dynamics, see [33]. The research strategy used was analytic auto-ethnography, an approach to research and writing that seeks to describe and systematically analyse (graphy) personal experience (auto) to understand cultural experience (ethno) [34].

Following [35], I collected data from my personal experiences as a consultant and director during RIDE from 2000 to 2013, with RIDE not taking place in 2004. In my role as director of several of the RIDE conferences, I kept process notes of my experiences during the different session. These process notes were used during the conference to make sense of the here and now, as well as there and then dynamics that occurred during the conference. After the conference these notes, my reflections and experiences were used as data in this study. I also used data from existing research (masters’ dissertations, articles, chapter, and conference presentations). From these different sources of data, I extricated particular vignettes relevant to the aim of this study. I used thematic analysis [33,36], from which themes emerged. I then used double hermeneutics [37] to interpret the data from the systems psychodynamic consultancy stance. The findings were integrated with relevant literature and different working hypotheses were generated [16]. Ethics clearance, 2015_CRERC_023(SD), was obtained for the research from the University of South Africa.

### 3.1. Findings

I have introduced RIDE to you as a container in which staff and members could study diversity dynamics. Now I would like to share with you my reflections and those of some of the staff members about the diversity dynamics that occurred in the RIDE society over 13 years, and consequently also in South African society. The two main themes are the diversity dynamics evident in earlier RIDE’s and diversity dynamics evident in recent RIDE’s.

### 3.2. Diversity Dynamics Evident in Earlier RIDEs

After the 2000 event, the members were asked about their experiences of RIDE. The following overview is based on these data obtained from members, and articles by [1,3,8], and my own reflections.

### 3.3. Race Used as a Container for Unresolved Diversity Matters

During first few RIDEs the RIDE citizens predominantly focused on the primary dimensions of diversity, with priority given to race and gender. As a participant stated: “It was as if this colour thing was still important for people to survive in the new South Africa”. This could be an indication of the extent to which the South African society fixated on race.

The struggle for power and position were evident in the early RIDE society—with power and status primarily allocated according to subgroup membership, especially with regard to race and gender.
The black male. The black men seemed to be empowered in the RIDE society. The older black men, who seemed to represent the struggle of the past, were very prominent at the beginning of the experience and became more silent as time went on in the early RIDEs. They appeared to contain the new stability in the country and “being there” may have been enough. The younger black men appeared to be very active and acted powerfully and assertively, with quite a lot of competition between them [1], p. 56.The black female. The older woman represented a mother figure to the group, i.e., the younger black people see her as a role model who looked after them during difficult times. The younger women were more silent than the younger black men but were very empowered. One black young woman expressed her need to take all opportunities, and for whites to get out of her way [1], p. 56.The coloured male. The older men seemed to be empowered and were apparently networking with everybody about working together in future. This could have enabled them not to take up the ascribed role projected onto them with regard to their racial representation. A coloured man stated: *“I don’t represent all the coloureds, but they expected me to carry this on behalf of all coloureds. The same happened with the Indians”* [1], p. 56.The coloured female. It appeared that coloured women had quite a difficult time relating to their racial representation within RIDE. They referred to “*struggling to find all of [their] parts*”, and their experience of being rejected based on the colour of their skin. A coloured woman stated: *“The RIDE once again made me aware of what I represent. It awakened a lot of feelings inside me. The most important was that childhood rejection of being coloured. It made me so angry, probably the most angry that I was in my entire life”* [1], p. 56.The Indian female. They seem to be caught between tradition and the new demands to be powerful and part of the new dynamic, expressing anger and carrying the pain of not belonging or being acceptable (not being black) in the new dispensation. An Indian woman stated: *“from day one I was being told that I am not black. I lived my whole life knowing that I am black”* [1], p. 56.The white male. Historically they were in power, the oppressors of others and were kept busy with the management of the country. It seemed as if they were unable to make contact with others, they appeared disempowered and often not heard by others. They seem to operate from the periphery. A white man stated: “*At a certain time in the programme I was really down and it felt as if there is no future for white (men) in the country*”. They also reported being pushed into offices at work which are out of reach from others and which make contact difficult. They seem to represent the shame of the past [1], p. 57. Two white men fell while walking during the first event—that could point to a lack of balance, where they felt disconnected from others, within the new South African dispensation [8]. Furthermore, it could represent an experience of their disempowerment in the new dispensation marked by the fall of apartheid.The white female. They seemed to have difficulty adapting to the new male role in the system. They were disillusioned towards white men and expressed their anger towards them for allowing the discrimination of the past. A white woman stated: *“Interesting for me was the anger I experienced against the white males who with their big mouths sat in the group and didn’t say a thing. Only afterwards they have a lot to say, but when they are back in the group they are silent. It is as if they are afraid of the black males”* [1], p. 57.

It seemed that the RIDE society was perpetuating a new dispensation with the difference that black males are now at the top. The RIDE society possibly did this in reaction to the apartheid dispensation with white males who were in power and other race groups having descending power and different control in relation to the white culture, as well as other cultures in the South African society. The message was clearly sent that they will not give up their newly found position on top of the ladder soon. A heated debate on the right to be called “black” or “African” vividly illustrated this struggle. Black participants heavily opposed the notion of white, coloured, or Indian people calling themselves African. By refusing other groups the right to be “African” or “black”, the black participants indirectly told the other groups that they would neither share their identity nor their position of power.

Although gender issues were also dealt with, gender played a secondary role and was seen to be only “women’s issues”. The men seemed to disown gender issues by projecting them onto the women. A black man stated: “What worried me (male) is that the gender issues, particularly the women have a lot of problems with it still.” The way the men disassociated themselves from the gender issues is not surprising, considering the idea that men have traditionally been the oppressor, while women have been the oppressed. The result is that only the oppressed are motivated to address issues relating to discrimination.

The notion that gender issues are women’s issues and should be worked on by women on behalf of the total system were further emphasised during the intergroup event, when an exclusive women’s group formed and found themselves sitting in the kitchen of the house used as one of the venues in a RIDE. I consulted this female group. Half of the women were black and half were white. After some discussion, two black women were sent out with a message to the larger system. Additionally, the next comment may be shocking to you, but as the black women were sent out I offered the interpretation based on what I saw unfolding in front of me that “Now the madams have sent the maids to do the work.” All the women were incensed by this interpretation and I was under verbal attack from them for making such an interpretation after they have worked so hard to complete the task of the intergroup. For me, it remained evident that the women’s group, instead of rallying together, had so much internal conflict and power struggles (race related) that their functioning was derailed. This may indicate that race-related issues in relation to who has the most power were once again predominant, even in the exclusive women’s group that wanted to rally together to fight for their cause. This reiterates the view that that South African society at that point of our journey found it difficult to move beyond race [3,9]. 

Based on the above discussion about how race was used in the RIDE society to position different groups, I hypothesise.

In the old dispensation a particular race hierarchy existed in the unconscious, i.e., white people being superior, with Indian, coloured, and black people carrying varying degrees of inferiority. In the new dispensation, we are possibly renegotiating the position of the race groups on a conscious and an unconscious level. These renegotiations may be attempts at reversing the roles among the race groups in the unconscious, i.e., black people being superior with Indian, coloured, and white people carrying varying degrees of inferiority. Although the socio-political dispensation has changed, in the South African psyche there is still investment about positioning different race groups in the unconscious in order for particular groups to be containers for our unacceptable parts [2]. In other words, before 2008, we South Africans may have been negotiating (in the unconscious) about which race groups would be the containers for our incompetence and inferiority. The aforementioned may also denote a shift in the power dynamics, i.e., the white males were in power in the previous dispensation, and in the recent RIDEs race and gender being used to understand the power of the different sub-groups in the new dispensation.

Please bear in mind that these diversity dynamics were evident at the beginning of the RIDE journey, probably in relation to the high-power distance in the South African culture. As the journey continued it seemed as if the RIDE citizens could move to other diversity dynamics, which pointed to a willingness to work with the complexity of diversity which was denied to South Africans during the old dispensation.

### 3.4. Diversity Dynamics Evident in Recent RIDEs

The following overview is based on a presentation by May, De Klerk, Motswoaledi, and Pretorius made at the annual conference of the Society for Industrial and Organisational Psychology in South Africa in 2013, as well as on Thabo Mofomme’s and my own subsequent reflections. In the following section, I will show how the RIDE society in more recent years has journeyed from focusing on race to other diversity preoccupations which may indicate a shift in apartheid power dynamics or alert us to power dynamics that we were not focusing on due to an initial pre-occupation with white males’ power in the apartheid dispensation.

### 3.5. Searching for the-Absent-White-Man

As fewer and fewer white men attended RIDE towards 2010, a new dynamic appeared, viz “looking for the white man.” The questions pertaining to “where is the white man?” or “why is the white man not here?” were asked on several occasions in the large study group (LSG) since 2010. All the participants and usually three consultants are present in the LSG studying the diversity dynamics of the entire system. It seemed that organisations mainly sent those who formed part of their employment equity committees to RIDE and somehow this meant that only a few white men attended. The dynamic is significant because in 2011 only one white woman, one coloured woman, and one Indian woman attended. In 2012, only three white women, one coloured woman, and one Indian woman attended. In 2013, there were no white women and no coloured or Indian person and no one asked: “where are they?” or “why are they not here?”. In a recent RIDE, when only one white man attended, during the intergroup and institutional event members formed groups called the elephants, the giraffes, and springboks. In the institutional event the springboks became “the mighty springboks”. Afterwards, discussing the dynamic “of the-absent-white-man” with a colleague, she commented “there he is—the mighty springbok”.

Thus, in the recent RIDE societies it appeared as if members replaced their “race preoccupation” by a preoccupation with “searching for the-absent-white-man.” This absence possibly caused confusion, because white men as our familiar receptacles for rage were not available to identify with these projections. It seems that “the-absent-white-man” is used as a trump card which RIDE citizens keenly played when they did not want to work with other challenging diversity dynamics that were present between themselves. Differently put, “the-absent-white-man” is a defended position used to skirt the responsibility of dealing with other diversities. I also hypothesise that this dynamic is an unconscious wish, to have a conversation with “the white-man-in-the-mind” and based on this conversation to explore whether a real connection, a good-enough relationship, can be formed with him. This unconscious wish for a good-enough relationship with the white man may contain our need to make good-enough relationships with those across difference. It is almost as if we have a fantasy that if we can manage the relationship with him, we can manage the relationship with anyone across difference.

### 3.6. Studying Leadership across and within Groups

In a more recent RIDE, during the sessions of the LSG, it seemed that the RIDE citizens wanted to explore leadership and diversity. The members studied female leadership across different race groups, male leadership within the same race group and leadership across different race groups and gender. This was evident from looking at whom the members placed at the beginning of the spiral. The chairs in the LSG are arranged in unusual configurations, usually a spiral—20 or more participants sit in a spiral. Please bear in mind that one of the assumptions for the LSG is that where you are placed in the spiral (where you sit in the LSG) is an unconscious group dynamic and has meaning that can be interpreted.

### 3.7. Engaging the Black Male Leader

In the LSG of a recent RIDE, it was evident that members wanted to study black male leadership, because they placed two black men at the beginning of the spiral. It seemed, to the consultants, as if the two black men represented Mandela and Zuma. I believe that members wanted to learn more about how these black leaders will treat and work with them—having messianic fantasies that these black male leaders will save them from the hard work that needs to be done, whether in the LSG or in South African organisations and society. Perhaps the members also wondered whether these leaders were good-enough, especially in the light of what we see in the exercising of leadership by black males in our daily lives. It can also be speculated that members felt some envy in the light of the mantle of leadership being bestowed upon the black man.

### 3.8. Male and Female Leadership across Different Race Groups

In another LSG session of the same RIDE, two of the consultants (a black woman and a white man) were placed in the first two chairs of the spiral. I believe that members wanted to learn more about how these leaders will treat and work with them—having messianic phantasies that the pair, across difference, will save them from the hard work that needs to be done whether in the LSG or in South African organisations and society. Perhaps they were also wondering what the pair, across difference, can create together, and whether they can work together and not be overwhelmed by conflict emanating from their differences.

### 3.9. Female Leadership across Race Groups

In the last LSG of the same RIDE, after we were seated, the consultants noticed that the first nine seats of the spiral were occupied by women—more or less in the following order: first chair a white woman, second chair a coloured woman, and third chair a black woman, who was a tea lady in her organisation. Bearing in mind that Julius Malema was reported in the media as referring to Lindiwe Mazibuko as a tea lady, it was clear that Zille, De Lille, and Mazibuko were seated in the first three chairs of the spiral. The next three seats seemed to represent the same three female leaders, as did the next three seats. We, the consultants, understood that the RIDE citizens wanted to study female leadership across race and age differences in South Africa by placing the three “DA female leaders” in the first nine seats of the spiral. Possible questions that we heard the RIDE citizens ask were:Are female leaders, especially black female leaders, good enough?Should the women not be adhering to traditional societal roles rather than interfering in the domain of men?How can women across race and age differences work together as leaders, and what kind of leadership can they offer us?

### 3.10. The Oppressor within: Exploring Diversity within the Same Race Group

A matter that has been skirted for quite some time by RIDE citizens is the diversity which exists within the same racial group. It was evident that members experienced difficulties in engaging with diversity within the same race group, especially members enjoying higher status in their organisations or society interacting with members occupying lower hierarchal levels in their respective organisations or societies. I received unusual evidence of this diversity dynamic during a recent RIDE. I was consulting to one of the groups in the intergroup session on day two and three—both sessions were just after lunch. In both sessions I was so sleepy that I almost fell asleep. It was so evident that I could not stay awake that a member of the group in another event referred to the fact that the consultant was sleeping in their session. I could not understand my experience because this has never happened to me before. Then, I received feedback in the staff meeting (based on work during a processing event) that one of the wealthier, successful black women reported that during the intergroup she felt that she had to look after and care for the lower status members on behalf of the group. Based on this experience the wealthier, successful black woman also reported that she could not connect authentically with black female members occupying lower positional levels in their respective organisations. It seems as if black female leaders/managers or those occupying higher echelon positions were acutely aware of their inability to relate to black women occupying lower positions and this created a somewhat debilitating effect on learning. There was some stuckness and not knowing how to handle this. It smacked of shame and guilt. This could also be a surprising discovery about our own resistance to change when we work with the unconscious, in that whereas we thought we were okay, it turns out we are not. This dynamic also alerted the staff about how we overlooked those (black) women occupying lower positions by keeping the language too technical for them. Once these (black) women could express themselves, they were quite actively involved in the process, suggesting that it was not inability or lack of interest on their part, but that we (the staff) also excluded them from participating.

In light of the above, I understood that this exploration of differences in racial sameness across position and socio-economic circumstances was the taboo in the room, and that it was so unspeakable that even I had to defend against it (by struggling not to fall asleep). Thus, in the more recent RIDE societies there may have been denial of discrimination based on status, with the refrain “*I don’t participate in status discrimination, it is others…*” This sounds like “*I am not a racist, it is others…*”.

Linked to this diversity dynamic of (not) dealing with diversity within racial sameness, was the difficulty for RIDE citizens to acknowledge and deal with other oppressions than that of white on black, e.g.,
The difficulty of black women in higher organisational positions to communicate with black women in lower organisational positions;The oppression of black women by black men;Conflict between different ethnic groups in South Africa.

By denying oppression amongst black people, discrimination is projected as only existing between races, with white people seen as the only discriminators. Through this projection, the black group can deny any conflict amongst them, preserve the apparent solidarity among themselves, and preserve themselves as good and white people as bad.

It is important to understand that this dynamic of denying diversity in the same race group does not only belong to the black group. The black group was merely used by the RIDE society to be containers for studying diversity within a group sharing a particular diversity characteristic.

### 3.11. Intersection between Race, Gender, and Class

In a recent RIDE, I experienced a situation which pointed to the intersection between diversity characteristics, viz, race, gender, and culture. In a session, an incident was related about how a person can make serious mistakes when unfamiliar with cultural traditions. Members coming from the same organisation, a black man, a white man, and a black woman, went to a meeting in rural Transkei. The villagers were waiting for them with the men seated on the one side and the women on the other. The white man wanted to go and sit where the women were seated, but the black man urgently gestured to him that he must join the men. I believe that the member related the incident, on behalf of all the participants, to communicate to me in particular and the other women in general that as a woman I should know my place and remain silent while the men attended to serious matters. I venture further to suggest that the unconscious wish among members could have been that my colleague Thabo Mofomme should be the director and not me.

In a later LSG, the group members started to speak in Sotho—it was a serious interchange which I did not understand. I was beginning to feel panicky because I could not assist my colleague, but I thought “Everything is in order. Because of his wealth of experience he will deal with the situation”. As I became more and more concerned because I was not able to make a useful contribution, a young Xhosa woman stood up and said somebody should translate because she did not understand what was going on. As you may realize, this was quite a relief to me and I thought, “Oh this is okay, I am not the only one who does not understand”. The next moment the black man at the beginning of the spiral left the room and the man on the second chair turned to whom we presumed was the oldest woman in the room to ask her about the role of women in the culture. She explained a woman’s role and then this was translated. The next moment my colleague turned to me and explained to me (and obviously to those who also did not understand) that this was a lekgotla, where the men could speak and the women had to remain silent. When the chief left (the man at the beginning of the spiral) the women were permitted to speak, but only after being asked by the motlatsa mookamedi (the second in charge), and then only the eldest woman in the group was permitted to speak. Based on what my colleague said, and the visual unfolding of the scenario, I understood and made the interpretation that based on culture, the women in the lekgotla had, in quite a sophisticated manner, been silenced by the men. It is significant to think that in this interaction the black group used the diversity characteristics of gender, culture, position, and age to silence 50% of the RIDE citizens in the LSG.

Based on the containment provided by my colleague and me, and the particular interpretation I made, it seemed that the work done by the men and the complicity of the women with the men in ensuring that the women remained silent could be seen as a diversity dynamic by the participants. This could have been a moment where the members could tolerate seeing an anxiety-provoking diversity dynamic because of being contained by RIDE and the consultants. Further evidence of this experience of containment came from a discussion about gender violence in the next LSG, where participants could acknowledge that the violence does not only entail men physically abusing women, but also women physically abusing men. In the conversation it felt as if men and women could hear each other differently or clearly across the gender divide—as if they could make space for each other’s pain and rage. I propose that perhaps these participants, through their experiences in this RIDE, developed their own capacity for reflecting on anxiety-provoking aspects [29,30] pertaining to some of their diversity dynamics.

## 4. Conclusions

In conclusion, I highlight a number of key insights or speculations gained from the shared reflections about diversity dynamics. These key points have been selected in the hope of thinking about how we can use this understanding in our organisations as educationists, managers, practitioners, and in society as citizens.

The above themes denote a journey of becoming aware of more diversity dynamics over a 13-year period of presenting RIDE first possibly as a shift in apartheid power relations and later focusing beyond this initial pre-occupation with white males’ power in the apartheid dispensation. It seems that through RIDE as a container, the consultants as containers, and the willingness of the participants to engage with their diversity challenges, more diversity dynamics have become accessible to the RIDE citizens. Through this we have become less stuck and less overwhelmed by a single (race) preoccupation. It also appears we have moved towards another pre-occupation, i.e., “the-absent-white-man” as a container for our wish to form good-enough relationships with those with whom we have difficulty connecting across difference. Although the black group seemed to be a container in recent RIDEs for working with diversity dynamics within a race group, this may denote an unconscious realisation that as South Africans we should also be working with differences in the same group. It also seems that we are exploring our relationships with leaders across and within (racial) differences. As the RIDE participants became more aware of these diversity dynamics, they could work more effectively with the marginalisation of groups through the intersection between diversity characteristics.

You may think that this is not a shift in the diversity dynamics, but an unhappy condition where we merely became aware of more diversity dynamics which we should work with. However, I hypothesise that, through the RIDE journey there appears to be a movement or shift from initial RIDEs in which participants projected the denigrated parts of themselves and their group into and onto other individuals and groups, to the more recent RIDEs where members seem to acknowledge and explore the denigrated parts in themselves and their own groups—perhaps to integrate the denigrated part with the idealised part within themselves. This seems to be a more layered and complex understanding of oneself and one’s diversity dynamics.

It is also important to realise that through RIDE as a container, the consultants as containers for more diversity dynamics have become accessible to the RIDE citizens and through this they have become less stuck and overwhelmed by a single race preoccupation. Perhaps the members have seen through the consultants’ calmness and self-reflecting stance in the presence of overwhelming rage and dread generated by the race preoccupation that they can tolerate the anxiety, pain, and rage elicited as they become aware of diversity dynamics and their complicity with these dynamics. I hypothesise that through experiencing RIDE and the consultants as good-enough containers the members develop their own capacity for reflecting on anxiety-provoking, enraging and distressing aspects [29,30] pertaining to diversity dynamics. In other words, by participating in RIDE, participants have moved from dumping their denigrated parts into and onto others, to reintegrating these parts with the idealised parts in themselves. Thus, hopefully moving beyond not merely spitting out/projecting into or onto others their unresolved diversity challenges. In other words, as South Africans we cannot get rid of our diversity dynamics, but we can learn to contain and integrate these diversity dynamics within ourselves and not dump our denigrated parts onto and into other South Africans as illustrated through Figure 2.

The importance of understanding the diversity dynamics operating in the relationship between individuals, between groups and in organisations in South Africa has been highlighted. In education, the understanding about diversity dynamics can be used by management to form a different awareness of conscious and unconscious factors impacting on the relationship between stakeholders and subsequently on the effectiveness of these stakeholders in their respective roles. This understanding about diversity dynamics also opens up the opportunity for stakeholders to explore how the different problems in the education landscape result not only from historical factors, but also from the diversity dynamics operating among themselves [2,8]. Other South African organisations should approach diversity not in a mechanistic manner, using instructional methods, such as lectures and presentations, but in a dynamic and experiential manner, such as using the group relations training model in interventions. Although participants in such events will experience resistance, opportunities for accepting personal responsibility for their positions and actions around diversity will be created [3]. Further, diversity interventions based on the group relations training model should be used in conjunction with other approaches, such as the socio-cognitive and legal imperatives currently used in organisations in order to optimise the management of diversity. It is thus not a case of opting for one or the other approach, but of using them together in order to gain a more comprehensive understanding of diversity, and therefore being able to manage more effectively.

I would like to return to the rock pile that was started by Mr. Mandela on Robben Island (Figure 3). I think of each one of the diversity dynamics that we now have access to as a rock that the RIDE citizens have placed on the rock pile. I imagine us now being able to place not only race but also gender, organisational position, socio-economic status and cultural group on this memorial and we build a society that is now free for the first time to really be cognisant of a range of diversity dynamics. I envisage that other South Africans through their relationship building across difference in communities, corporate life, the public sector, educational institutions, spaces such as RIDE, and in the different spheres of their daily lives will add to our understanding of and access to our diversity dynamics. Drawing on the symbolic value of building a rock pile while journeying, I suggest that the RIDE citizens have assisted us by starting a rock pile that we can all add to as we learn about our own diversity dynamics and work with these in the South Africa we find ourselves in every day. Considering this rock pile and the work that has been done before us, this rock pile will assist us not to lose our way as we learn to build relationships across difference. Furthermore, through this rock pile made up of South African diversity dynamics we acknowledge the sweat and toil by those during RIDE and other South Africans and we draw on what they have learnt to continue our journey of dealing with diversity challenges in our efforts to become an integrated nation

## Figures and Tables

**Figure 1 ijerph-19-09064-f001:**
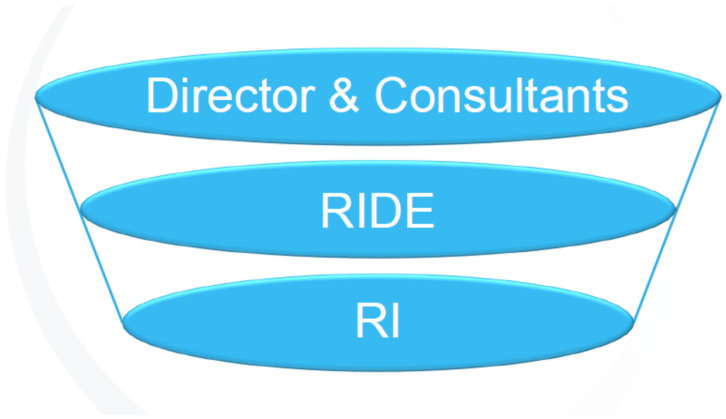
Containment of diversity dynamics.

**Figure 2 ijerph-19-09064-f002:**
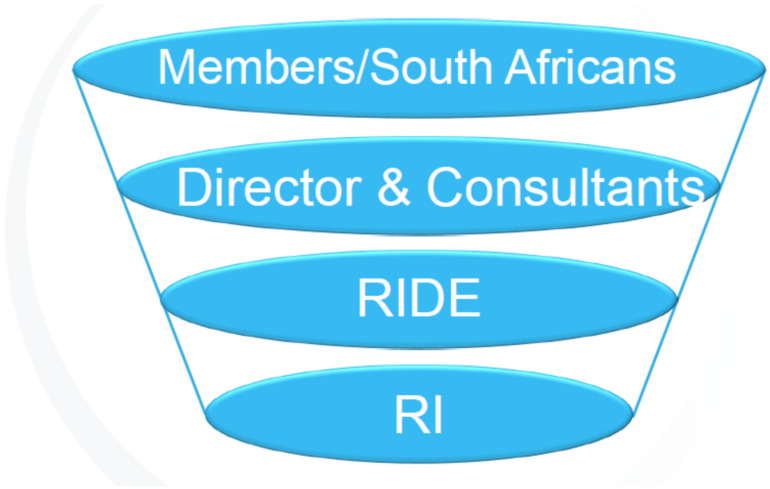
Containment of diversity dynamics expanded.

**Figure 3 ijerph-19-09064-f003:**
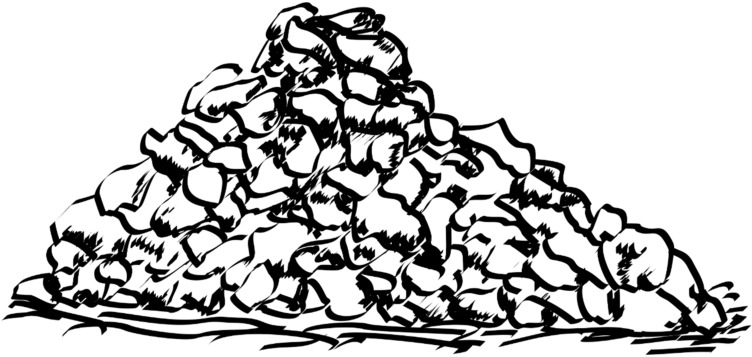
Pile of rocks representing the rock pile started by Nelson Mandela and added to by former prisoners of Robben Island Prison, South Africa.

## Data Availability

Data formed part of author’s process notes and has been stored as part of her consultancy notes in accordance with the ethical requirements of the Health Professions Council of South Africa.

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
