# Peer review of "The Robben Island Diversity Experience: It’s about the Journey, Not the RIDE"

_ijerph, 2022, doi:10.3390/ijerph19159064_

Round 1

Reviewer 1 Report

While I don't have much experience reviewing autoethnographies, I have published several qualitative articles and I have taught courses in this area.  I certainly appreciate this article and find it very interesting, as will many readers of the journal.  Thus, I think it should be published. 

However, I have several suggestions that might make the analysis of the RIDE program more understandable to readers.  Basically, I am recommending more information related to the context of the program that also includes more information about the author.  

First, I think it would be useful for the author to provide more information about her background.  There are bits and pieces of information scattered about in the article, but bringing it together in a paragraph in the introduction would help the readers better understand the author's perspective and potential biases.  Since race and gender are important parts of the article, perhaps the author could share her experiences about these issues .  As an autoethnography, we need to know more about the author.  

Second, it would be useful to provide a review of programs that seek to accomplish similar goals to the RIDE program.  Are there other programs in Africa or in other countries that seek to accomplish this kind of diversity training in this kind of context?  Google Scholar brings up many articles and books about effective diversity training, and a brief review of the most cited articles might be useful to place the RIDE program in context.

Third, the article doesn't tell us much about the high power distance South African culture.  The old culture that appears to be confronted by the program placed white males at the top with other groups have descending power in a specific order.  If this is the case, it would be useful to talk about South African culture focusing on the dimensions relative to the analysis, which seems to be power.   

This leads me to my final comment related to the goal of the program.  The goal of the program as articulated on page 4 (Primary task of RIDE) is very ambiguous.  Is the real goal of the program to confront the old South African high-power distance culture and replace it with a culture that is less hierarchical related to group dynamics?  What does the goal of "working on their own diversity dynamics? mean?  What's the real goal of the program and what does the program want the participants to leave with?  

My goal in this review is to provide some questions for the authors to think about that readers might be having as they read the article.  Give us more context for the analysis so we can really understand the RIDE program.  

Author Response

Thank you for the review of the article. I have addressed the issues raised to the best of my ability.

Introdution to the article has been reformulated.

Although it may come across as such, this is not an analsyis of the RIDE programme. This is rather my reflections about the aspects about diversity dynamics that the particpants have been raising from 2000 to 2013 and a shift with regards to the diversity dynamics during the RIDE programme. Further, information about RIDE was provided under the headings:

  • RIDE: Container of diversity dynamics
  • RIDE: Group Relations Training event
  • Primary task of RIDE
  • Resourcees
  • The director duirng the Group Relations event
  • Consultants during the group relations training event

I am left with the sense that I do not quite understand the reviewer and will appreciate more feedback about whta should be expected.

I have included some information about my background as required in an  autoethnographic study. 

Review of programs that seek to accomplish similar goals to the RIDE program: Refered to other research on two conferences which could be similar to RIDE in the introduction.

High power distance South African culture: I have referred to thsi matter under teh subheadings Race used as a container for unresolved diversity matters,  Diversity dynamics evident in recent RIDEs and the conclusion.

The goal of the program vs primary task: I did not change the primary task of RIDE because this is indeed the task that we worked towards in teh role of director, consultants and the participants. In other words each perons has their onw iversity dynamics based on teh repsensetarion in the world. Through RIDE each particpant could work on tehir diversity dynamics as shaped by the intersectionality of tehir diversity characteristics. In other words, each participant will learn something about themeslves in relation to the intersectionality of their diversity characteristics and how this impact their relationships with others

Reviewer 2 Report

Overall, this is a very good research paper. It is also timely because countries that have practised apartheid as a system of government eventually have had to come to terms with diversity. The United States, an older democracy, is still grappling with diversity; South Africa is still a relatively young country as a multiracial democracy. 

It is good that the diversity conferences (I would have preferred the term "workshops") are taking place in Robben Island, a perfect symbol of the destructive nature of apartheid. Your introduction, in which you describe Robben Island and what it stands for, is perfect. I also like your writing style, especially your ability to return to the powerful symbol of the rock piles in your conclusion.

However, there are a few minor things you should take care of:

1. lines 175-177: Explain or correct the seemingly time contradiction involved between when you were the director and Motswoaledi's. If you were RIDE director from 2002-2013, how could Motswoaledi be the director in 2006? 

2. line 249: "FROM 2000 TO 2013". Use lower case instead of upper case letters.

3. line 313: Change "form" to "from".

4. line 347: Change "venue" to "venues".

5. line 350: Change "send" to "sent".

6. line 452: Change "was" to "were".

7. line 559: Change "though" to "through". 

8. line 568: Delete the second "that".

9. line 579: Add "of" after "aware" to read "aware of".

10. line 580: Change "though" to "through".

11. Make lines 582-591 the same paragraph instead of two paragraphs, and edit it appropriately to be so. 

12. line 602: Delete comma after "their," to make it "their".

13. line 618: Change "from from" to "from".

14. line 651: Delete hyphen from "Pris-on" to become "Prison".

15. lines 675-679: The reference entry on Lazar, R.A. should come after Koortzen, P. to follow alphabetical order of referencing. 

16. line 701: Give a space before "Proceedings" and another before "Jupiter."

Author Response

Thank you very much for the review I have attended to the changes as requested,  I will also request editing of the document.

Diversity conferences (I would have preferred the term "workshops"): I understand how in the description of RIDE the reviewer would have preferred the term "workshops." Following the tradition of events using the goup relations training model, I will refer to RIDE as a conference.  

lines 175-177: Explain or correct the seemingly time contradiction involved between when you were the director and Motswoaledi's. If you were RIDE director from 2002-2013, how could Motswoaledi be the director in 2006?  Matter addressed

All other formatting issues addressed.

I will also request editing of the document.

Reviewer 3 Report

1.  Abstract should include brief summary of introducing for the topic. Then it should continue with summary of method, finding and implications of the study. Abstract should be written again by taken aforementioned points. Author did not mention sample size of the study, results and implications.

2. Introduction part should give general information of the topic then purpose of the study. However, author directly started with the purpose of the study. On the other hand, purpose of study is “ describe the experiences of a few hundred people participating in the Robben Island 30 Diversity Experience (known as RIDE) from 2000 to 2013, using analytic autoethnogra- 31 phy to analyse and interpret the conscious and unconscious diversity dynamics operate- 32 ing among these participants “  or “explore the extent to which we place our denigrated 41 or unacceptable parts into other individuals and groups, and to show how the group 42 relations training model is a way of learning to take back and integrate these exported parts into ourselves in order to have a more layered and complex understanding of diversity in ourselves, our organisations and South African society”. Please make it clear or make a difference what is the objective of the study and what you are seeking to answer?

3.  Please open a Title as a Literature Review and use Robben Island: Container for diversity and others until Methodology as a part of the Literature review. This way will be  more appropriate.

4. After the Findings please make it clear that which title also belongs to findings.

5. Its advisable to use table for the questions you asked.

6. Conclusion is sufficiently written and future direction of the study is mentioned.

7. Reference list is a bit less but also there is some mistake regarding to style. For example, in the some reference page number shown by using (pp) and in some only number used. So please check the references again and make sure same style u have used for all.

Author Response

Thank you for the review of the article. I have addressed the issues raised to the best of my ability.

1. Abstract was rewritten.

2. Introduction re-written. The aim of the article stated at the end of the introduction.

3. Heading Literature included and use Robben Island: Container for diversity and others until Methodology as a part of the Literature review. 

4. Addressed matter of which subheadings were part of the Findings.

5. I was unsure how to address the issue of table for the questions. However, I prefer the questons as part of the discussion and not to foreground the questions in a table. In this way the different aspects of the findings have similar importance.